

# Low-dose lipopolysaccharide inhibits spinal cord injury-induced neuronal apoptosis by regulating autophagy through the lncRNA MALAT1/Nrf2 axis

Jianhua Hu[1,2,*], Kun Huang[1,*], Feilong Bao[2,3], Shixiao Zhong[2,3], Qianbo Fan[2,3] and Weichao Li[1,2,3]

[1] Orthopedic Surgery, The First People's Hospital of Yunnan Province, Affiliated Hospital of Kunming University of Science and Technology, Kunming, China
[2] Faculty of Medical Science, Kunming University of Science and Technology, Kunming, China
[3] Yunnan Key Laboratory of Digital Orthopaedics, Kunming, China
[*] These authors contributed equally to this work.

Corresponding author
Weichao Li, liweichao0394@126.com

## ABSTRACT

**Background.** Spinal cord injury (SCI) is a neurological disease associated with a high disability rate. Low-dose lipopolysaccharide (LPS) has been reported to activate cross-immune tolerance and alleviate the effects of various traumatic stimuli. The present study aimed to explore the effect of LPS on SCI and the potential molecular mechanism.
**Methods.** Male Sprague-Dawley (SD) rats were used to established an *in vivo* SCI model and were intraperitoneally injected with lentivirus particles encoding a MALAT1 small interfering RNA (siRNA) on day 10 prior to SCI and with 0.2 mg/kg LPS 72 h prior to SCI. Basso, Beattie, and Bresnahan (BBB) scoring; HE staining; and TUNEL assay were used to assess neurological function and pathophysiological changes. Western blot and immunohistochemistry (IHC) were used to detect cell autophagy and Nrf2 nuclear translocation. PC12 cells were exposed to oxygen-glucose deprivation/reoxygenation (OGD/R) to establish an *in vitro* SCI model. *In vitro* SCI model cells were pretreated with LPS and transfected with siMALAT1 or MALAT1 overexpression plasmid aimed at knocking down MALAT1 or overexpressing MALAT1. The cell counting kit-8 (CCK-8) assay was used to measure the toxicity of LPS towards PC12 cells. Flow cytometry and immunofluorescence analysis were performed to investigate cell apoptosis and Nrf2 nuclear translocation.
**Results.** SCI rats preconditioned with low-dose LPS had higher BBB scores, reduced SCI injury, increased MALAT1 expression and activated autophagy and Nrf2 nuclear translocation in the *in vivo* SCI model. In the *in vitro* SCI model, low-dose LPS treatment suppressed the apoptotic ratio of PC12 cells, increased MALAT1 expression, activated autophagy, and promoted Nrf2 nuclear translocation. Silencing MALAT1 exacerbated OGD/R injury *in vitro* and weakened the protective effect of low-dose LPS. Overexpression of MALAT1 inhibits OGD/R-induced apoptosis by inducing autophagy and promoting Nrf2 nuclear translocation. This was also been confirmed in animal experiments, silencing MALAT1 blocked the promotion of Nrf2 by low-dose LPS and the alleviated of SCI apoptosis.
**Conclusions.** Low-dose LPS exhibited a protective role on SCI by activating autophagy and suppressing nerve cell apoptosis via the lncRNA MALAT1/Nrf2 axis.

## INTRODUCTION

Spinal cord injury (SCI) is regarded as a devastating neurological disease. Generally, the primary cause of SCI is traumatic events, such as sudden falls, sports injuries, and traffic accidents (*Cowan, Lakra & Desai, 2020*). The incidence of SCI grows each year, with traffic accidents and falls accounting for more than 80% cases (*Hamid et al., 2018*; *Montoto-Marqués et al., 2017*). SCI causes severe sensory, motor, and autonomic nerve dysfunctions, affecting the physical and mental health of patients (*Ahuja et al., 2020*). At present, clinical treatments for SCI mainly focus on surgical decompression, neurotrophic drug support, the prevention and treatment of complications, and functional rehabilitation (*Ahuja et al., 2017*; *Holmes, 2017*). However, the neurological function recovery of patients with SCI resulting from these treatments remains unsatisfactory. It is thus vital to understand the molecular mechanism underlying SCI pathogenesis to obtain a direction for SCI therapy development.

Initial mechanical injury leading to SCI is usually followed by secondary injury, such as through an inflammatory response or oxidative stress; this leads to damage of the spinal canal neural structure and, thereby, aggravates SCI (*Anjum et al., 2020*; *Zhaohui & Shuihua, 2020*). Apoptosis of spinal nerve cells caused by a series of immune-inflammatory responses is considered to be the key cause of SCI aggravation (*Rong et al., 2022*). Autophagy is a vital physiological process required by cells to maintain the stability of their internal environment (*Mizushima & Levine, 2020*); it participates in various cellular processes, such as the regulation of inflammatory responses in neurodegenerative diseases, cancers, and cardiovascular diseases (*Yao et al., 2021*). Defects in autophagy can disrupt protein homeostasis; lead to nerve cell death, axonal degeneration, and synaptic dysfunction; and, thereby, induce neurodegenerative diseases (*Mizushima & Komatsu, 2011*). Due to the accumulation of damaged organelles and proteins in nerve cells during SCI, the activation of autophagy is essential to protect nerve cells. *Zhou et al. (2020)* suggested that the inhibition of TFE3 leads to a decrease in autophagy flux and subsequent ER stress-induced apoptosis in neurons, ultimately impeding functional recovery after SCI. However, accumulating evidence also shows that the over-activation of autophagy not only fails to play a role in cell protection but also leads to excessive self-digestion, resulting in programmed neuronal cell death. For example, *Yin et al. (2019)* found that angiopoietin-1 alleviated SCI by inhibiting autophagy in rats. Therefore, autophagy plays an important role in SCI, but it is unclear whether this role is neuroprotective or neurodegenerative.

Long non-coding RNAs (lncRNAs) are a kind of functional transcripts with a length of more than 200 nucleotides (*Bridges, Daulagala & Kourtidis, 2021*). More and more studies are showing that lncRNAs take part in the regulation of the inflammatory response and neuronal activity after SCI (*Cheng et al., 2020*; *Wang et al., 2021*; *Zhang et al., 2021*). Metastasis-associated lung adenocarcinoma transcript 1 (MALAT1) has been shown to perform important regulatory functions in various cellular activities, including cancers,

vascular diseases, and neurological diseases (*Qiao, Tu & Liu, 2021*; *Zhang et al., 2017b*; *Zhu et al., 2019*). A recent study showed that MALAT1 exerted a neuroprotective effect in a rat model of SCI (*Qiao et al., 2018*). Moreover, *Zhu et al. (2020)* found that MALAT1 suppressed hypoxia/reoxygenation-induced human umbilical vein endothelial cell injury. Moreover, MALAT1 was reported to participate in regulating autophagy (*Yang et al., 2020*; *Zhu et al., 2019*). For instance, *Wang et al. (2019a)* found that MALAT1 activated autophagy in OGD-induced brain microvascular endothelial cells. However, how MALAT1 affects SCI progression by regulating autophagy is unclear.

Lipopolysaccharide (LPS) is considered to be the main cause of systemic inflammatory syndrome (*Chakraborty & Francis, 2020*; *Zamyatina & Heine, 2020*). Recent studies have shown that LPS induces a strong immune response after interacting with its receptors on cell membranes (*Rathinam, Zhao & Shao, 2019*; *Yoshida et al., 2018*). It is worth noting that low-dose LPS activates cross-immune tolerance and alleviates various traumatic stimuli in Swiss albino mice (*Jacob et al., 2016*). Evidence has indicated that low-dose LPS pretreatment exerts protective effects on injuries of the liver, kidney, intestines, and brain (*Chang et al., 2014*; *He, Xia & Zhang, 2018*; *Nakasone et al., 2016*; *Wang et al., 2017*). Moreover, several studies have indicated that the therapeutic role of LPS pretreatment on SCI might be mediated by the regulation of the inflammatory response and suppression of neuronal apoptosis (*Hayakawa et al., 2014*; *Li et al., 2016*; *Li et al., 2014*). Studies have also shown that low-dose LPS pretreatment activates nuclear factor E2-related factor 2 (Nrf2) and inhibits the inflammatory response in SCI (*Zhu et al., 2015*). A recent study reported that LPS induced the lncRNA MALAT1 expression to regulate inflammatory cytokine production (*Nan et al., 2020*). However, the specific mechanism through which LPS regulates MALAT1 and autophagy to prevent SCI progression is not fully understood.

The present study aimed to systematically clarify the roles of low-dose LPS in SCI and the potential mechanisms based on *in vivo* and *in vitro* experiments. The findings of this work might provide new insights and strategies for the treatment of SCI in the future.

# METHODS

## Animal studies

The clean-class seventy-two male Sprague-Dawley (SD) rats (2.0−2.5 months old, 250–300 g) were obtained from Shanghai SLAC Laboratory Animal Co. Ltd (Shanghai, China). All rats were housed with sterile food and drinking water at a temperature of 20 °C to 24 °C, humidity of 50% to 60%, 60 air exchanges per hour in the cages, and a 12 h light/dark cycle.

The *in vivo* SCI model was established as previously described (*Li et al., 2018*). In brief, rats were anesthetized by inhalation of 2% isoflurane (Kermel, Tianjing, China). After anesthesia administration, each rat was placed in the prone position and fixed on a plate. Sterile ophthalmic scissors were used to incise the skin to expose the T9/T10 vertebrae. Allen's weight drop apparatus was used to drop a weight of 8 g from a 40-mm height onto the dural sac in order to cause moderate SCI damage ("trauma hit" injury). The injured spinal cord was then rinsed with saline (Beyotime, Jiangsu, China), and the incision was

sutured. The bladders of the rats were massaged twice daily to promote urination and prevent infection until spontaneous urination resumed. Successful establishment of the *in vivo* SCI model was based on the appearance of the following symptoms: tail with spastic swing, spinal cord edema and hemorrhage, and retraction-like flutter and flaccid paralysis of bilateral hind limbs.

To explore the effect of low-dose LPS on SCI *in vivo*, the model rats were randomly divided into three groups: (1) Sham group ($n = 8$), (2) SCI group ($n = 8$), (3) SCI+LPS group ($n = 8$). Rats in the SCI+LPS group were intraperitoneally injected with 0.2 mg/kg LPS (Sigma, USA) 72 h prior to injury (*Li et al., 2014*). Rats in the SCI group were intraperitoneally injected with 0.2 mg/kg saline (Beyotime, China) 72 h prior to the injury. Rats in the Sham group were only given a laminectomy without trauma hit and received 0.2 mg/kg saline.

To determine the effect of MALAT1 on SCI *in vivo*, model rats were randomly divided into four groups ($n = 12$ each): (1) Sham group (given laminectomy without trauma hit; intraperitoneally injected with 0.2 mg/kg saline), (2) SCI+siNC group (intraperitoneally injected with lentivirus particles encoding a negative control siRNA on day 10 prior to injury; injected with 0.2 mg/kg saline 72 h prior to the injury), (3) SCI+LPS+siNC group (intraperitoneally injected with lentivirus particles encoding a a negative control siRNA on day 10 prior to injury; injected with 0.2 mg/kg LPS 72 h prior to injury), (4) SCI+LPS+siMALAT1 group (intraperitoneally injected with lentivirus particles encoding MALAT1-siRNA on day 10 prior to injury; injected with 0.2 mg/kg LPS 72 h prior to the injury).

Both the research team and the veterinary staff monitored animals twice daily. Health was monitored by weight, food and water intake, and general assessment of animal activity. On day seven post-injury, all rats were were sacrificed using $CO_2$ inhalation, and the spinal cord tissues were collected and stored at −80 °C for further experiments. The following parameters were assessed: Basso, Beattie, and Bresnahan (BBB) locomotor rating scale; hematoxylin-eosin (HE) staining; terminal deoxynucleotidyl transferase (TdT) dUTP Nick-End Labeling (TUNEL) assay.

Animal experiments were approved by the Ethics Committee of The First People's Hospital of Yunnan province (Affiliated Hospital of Kunming University of Science and Technology) (NO.: KMUST-MEC-131), and in line with the Guide to Animal Care and Use of The First People's Hospital of Yunnan province (Affiliated Hospital of Kunming University of Science and Technology).

## Assessment of neurologic function

Neurologic function after SCI was assessed using the BBB locomotor rating scale (*Basso, Beattie & Bresnahan, 1995*). This scale is based on the observation of ankle joint movement, gait coordination, and parallel paw placement. Rats in each group were evaluated at 12, 24, 36, and 48 h post-injury. The rats were made to empty their bladders before scoring to avoid extraneous effects on the results. The tests were independently performed by three non-professional personnel familiar with the scoring criteria; the average of the scores given by these three persons was finally used.

### HE staining

The spinal cords tissues were fixed with 4% paraformaldehyde (Beyotime, Jiangsu, China) for 48 h at 25 °C. Then, the tissues were dehydrated in an ascending gradient of ethanol and then made transparent with xylene. Next, the tissues were embedded in paraffin wax and sliced into 4-$\mu$m-thick sections. Afterward, the sections were dyed with hematoxylin for 2 min and eosin for 1 min at 25 °C. Images of stained sections were viewed by a light microscope (CH30, Olympus, Japan).

### TUNEL assay

The cell apoptosis was analyzed by a TUNEL Assay Kit (Roche, Indianapolis, IN, USA). In brief, the sections were treated with proteinase K for 15 min and 3% hydrogen peroxide) for 20 min at 25 °C. Then, the sections were incubated with 50 $\mu$L TUNEL mixture for 1 h and immersed by diaminobenzidine for 2 min at 25 °C. Images of stained sections were viewed by a fluorescence microscope (OLYMPUS, Tokyo, Japan).

### Immunohistochemistry (IHC) staining

The sections were dewaxed and treated with 3% $H_2O_2$ to deactivate endogenous peroxidase, and added 5% bovine serum albumin for 1 h at 25 °C to block non-specific antigen. Then, the sections were incubated with anti-Nrf2 antibody (ab31163, 1:100 dilution, Abcam) at 4 °C overnight and the secondary antibody (ab7092, 1:500 dilution, Abcam) for 1 h at 25 °C. Afterward, the sections were stained with DAB (Beyotime) for 1 min. Images of stained sections were viewed by a light microscope (CH30, Olympus, Japan).

### Cell culture and cell treatment

The rat adrenal chromaffin cell tumor differentiation cell line, PC12 (ATCC, USA) was cultured in Dulbecco's Modified Eagle's Medium (Sigma-Aldrich, St. Louis, MO, USA) containing 10% fetal bovine serum (Hyclone, Logan, UT, USA) under humid conditions of 37 °C and 5% $CO_2$. The PC12 cells were pretreated with or without 0.05, 0.10, 0.15, 0.20, 0.30, and 0.40 $\mu$g/mL lipopolysaccharide (LPS; Sigma-Aldrich, St. Louis, MO, USA) for 24 h at 37 °C. Then, these cells were exposed to oxygen-glucose deprivation/reoxygenation (OGD/R) to simulate an *in vitro* SCI model, as previously described (*Bao et al., 2016*). For OGD/R, briefly, PC12 cells were first exposed to OGD for 4 h and then treated with reoxygenation (R) for 24 h.

### Cell transfection

To knock down MALAT1 expression, the sequences of siRNA against MALAT1 (siMALAT1) and siNC were synthesized by Genepharma (Shanghai, China). siMALAT1: 5′-ATGAGAAGAACCAAGTTTTAACC-3′; siNC: 5′-TTCTCCGAACGTGTCACGT-3′. To overexpress MALAT1 expression, the cDNA of MALAT1 was cloned into pcDNA−3.1 vector. For the transfection, PC12 cells were transfected with 50 nM of siMALAT1 or MALAT1 overexpression plasmid when the degree of cell fusion reached 70% using the Lipofectamine 3000 reagent (Invitrogen, Carlsbad, CA, USA) for 24 h at 37 °C following with the manufacturer's instructions.

## Cell viability assay

The cell viability of PC12 cells was detected by the CCK-8 Kit (Beyotime, China). In brief, PC12 cells were plated into 96-well plates with $1 \times 10^4$ cells/well for 24 h. After OGD/R and transfection, 10 µL CCK-8 reagent was incubated with PC12 cells for 1 h at 37 °C. Then, the optical density was detected by a microplate reader (Biotek, Winooski, VT, USA) at 450 nm.

## Cell apoptosis assay

The cell apoptosis of PC12 cells was detected by the Annexin V-FITC Apoptosis Detection Kit (BD Biosciences, San Jose, CA, USA). Briefly, PC12 cells were resuspended in Binding Buffer. Then, 500 µL cell suspension was incubated with 5 µL Annexin V-FITC and 10 µL PI for 10 min. The incubated cells were protected from light. At last, the apoptotic rate was assessed by a flow cytometer (BD Biosciences, San Jose, CA, USA).

## Immunofluorescence staining

PC12 cells were fixed with 4% paraformaldehyde for 15 min and blocked with serum for 30 min. Then, the cells were incubated with primary antibody anti-Nrf2 (ab31163, 1: 100 dilution, Abcam) overnight at 4 °C and the fluorescent secondary antibody (ab7092, 1:500 dilution, Abcam) for 90 min at 37 °C. Following, the nuclei were stained with DAPI for 5 min at 25 °C. The images were observed by a fluorescence microscope (OLYMPUS, Tokyo, Japan).

## Reverse transcription-quantitative PCR (RT-qPCR)

Total RNA from tissues or cells was isolated using Trizol (Takara Biotech Japan) and evaluated by NanoDrop ND-1000 (Thermo Fisher Scientific, Waltham, MA, USA). Then, total RNA was transcribed into cDNA using a PrimeScript RT Kit (Takara Biotech, Japan) at 37 °C for 15 min and 85 °C for 5 s. RT-qPCR was performed by ABI Prism 7500 PCR system and SYBR Green Master Mix (Applied Biosystems, Foster City, CA, USA). The relative expression levels were calculated by $2^{-\Delta\Delta CT}$. GAPDH was used as an internal control. The primer sequences were listed: MALAT1, forward: 5′-ATGCGAGTTGTTCTCCGTCT-3′, reverse: 5′-TATCTGCGGTTTCCTCAAGC-3′; GAPDH, forward: 5′-GGAGTCCACTGGTGTCTTCA-3′, reverse: 5′-GGGAACTGAG CAATTGGTGG-3′.

## Western blot

Total protein from tumor tissues or cells was extracted by RIPA buffer (Beyotime, Jiangsu, China). A Nuclear and Cytoplasmic Protein Extraction Kit (Beyotime) was used to isolate nuclear protein and cytoplasmic protein. All isolated protein was quantified by a BCA Protein Assay Kit (Boster Biological Technology, Wuhan, China). A total of 40 µg protein samples were separated in 10% sodium dodecyl sulfate-polyacrylamide gel electrophoresis, and transferred to PVDF membranes. After blocking with 5% skim milk, the membranes were incubated with the primary antibodies at 4 °C overnight, and secondary antibody (ab7092, 1:2000 dilution, Abcam) for 1 h at 25 °C. At last, the ECL Detection Reagent (Thermo Fisher Scientific, Waltham, MA, USA) was used to visualize the membrane. The

protein band analysis was conducted with ImageJ software. The antibodies used were listed as follows: anti-Beclin1 (ab207612, 1:2000 dilution, Abcam); anti-LC3II/LC3I (ab192890, 1:2000 dilution, Abcam); Nrf2 (ab92946, 1:1000 dilution, Abcam); anti-Histone H3 (ab32356, 1:10000 dilution, Abcam); anti-GAPDH (ab181602, 1:10 000 dilution, Abcam).

## Statistical analysis

All experiments were repeated three times independently. The data were analyzed by GraphPad (6.0 version) and presented as the mean $\pm$ standard deviation. Two-tailed Student's $t$-test and one-way ANOVA were performed to compare two and multiple groups, separately. $P$ value $< 0.05$ is viewed as statistically significant.

## RESULTS

### Low-dose LPS alleviates neurologic dysfunctions and histopathological damage in the *in vivo* SCI model

To explore the effect of low-dose LPS on SCI, SD rats were intraperitoneally injected with 0.2 mg/kg LPS 72 h prior to SCI. BBB scores were used to analyze the neurologic function after SCI. Compared to rats in the Sham group, those in the SCI group had lower BBB scores at 12, 24, 36, and 48 h ($p < 0.001$); however, LPS pretreatment partly recovered the decreased BBB scores caused by SCI ($p < 0.01$) (Fig. 1A). Next, pathological and morphological injuries of the spinal cord were detected using HE staining. The spinal cord tissue structure in the Sham group was intact, without necrotizing cells and inflammatory cell infiltration, while there was severe structural injury of the spinal cord tissue, with a necrotizing cystic cavity at the injury center in the SCI group; LPS pretreatment effectively alleviated the structural injury and inflammatory cell infiltration in the SCI group (Fig. 1B). Furthermore, TUNEL assay showed that the cell apoptosis rate was much higher in the SCI group compared to that in the Sham group ($p < 0.001$) and that LPS pretreatment significantly reduced the cell apoptosis rate in the SCI group ($p < 0.01$; Figs. 1C, 1D). Overall, these results indicate that low-dose LPS has protective effects on the recovery of nerve function and pathophysiological changes in our rat model of SCI.

### Low-dose LPS increases MALAT1 expression, activates autophagy, and induces Nrf2 nuclear translocation in the *in vivo* SCI model

It has been reported that MALAT1 is associated with SCI progression (*Zhou et al., 2018*). Thus, we next assessed MALAT1 expression in spinal cord tissues. Compared to that in the Sham group, the level of MALAT1 was observably increased in the SCI group ($p < 0.05$); LPS pretreatment further increased the MALAT1 expression in the SCI group ($p < 0.01$; Fig. 2A). Next, the autophagy-related proteins Beclin1 and LC3II/LC3I were analyzed using Western blot. The expression levels of Beclin1 and LC3II/LC3I were slightly increased in the SCI group compared to those in the Sham group ($p < 0.05$); LPS pretreatment dramatically enhanced the Beclin1 and LC3II/LC3I expression levels in the SCI group ($p < 0.05$, $p < 0.001$; Figs. 2B–2C). IHC staining showed that, compared to the Sham group, there was higher activation of Nrf2 and induction of Nrf2 nuclear translocation in the SCI group ($p < 0.01$); LPS pretreatment further enhanced Nrf2 nuclear translocation

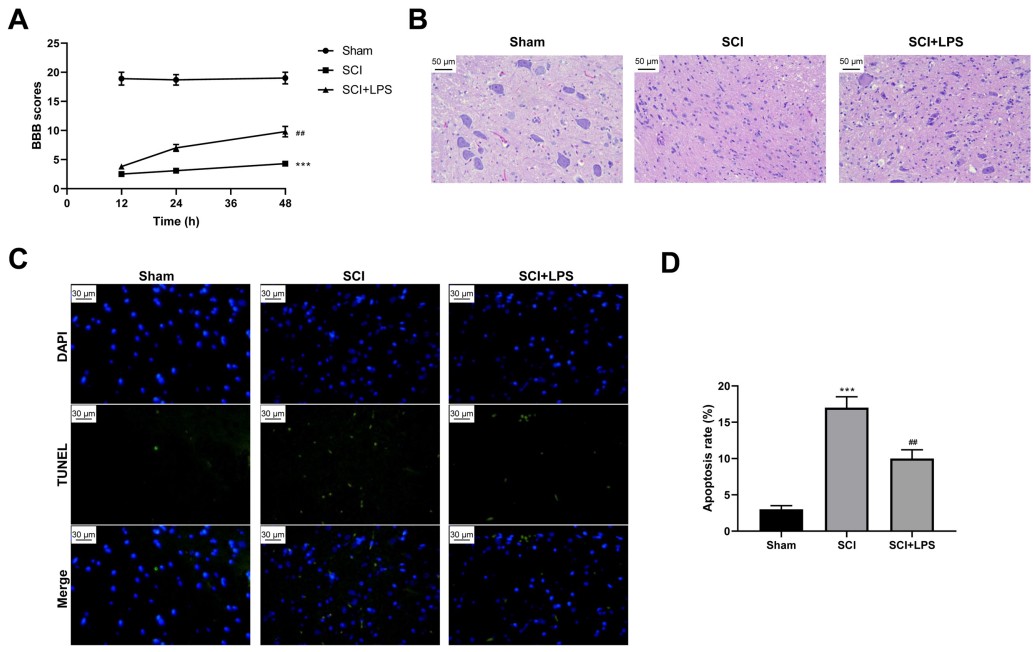

**Figure 1** **Low-dose LPS alleviates neurologic dysfunctions and histopathological damage in the *in vivo* SCI model.** SD rats were intraperitoneally injected with 0.2 mg/kg LPS 72 h prior to SCI. (A) BBB scores were used to analyze the neurologic function after SCI. (B) The pathological and morphologic injury of spinal cord tissues was observed by HE staining. Scale Bar = 50 μm. (C, D) TUNEL assay was performed to observe the cell apoptosis in spinal cord tissues. Scale Bar = 50 μm. $n = 8$ each group. *** $P < 0.001$ *vs.* Sham; ## $P < 0.01$ *vs.* SCI.

in the SCI group ($p < 0.05$; Figs. 2D, 2E). All these data suggest that the protective effect of low-dose LPS on SCI may be attributed to the increased expression of MALAT1, autophagy activation, and Nrf2 nuclear translocation.

## Low-dose LPS increases MALAT1 expression, activates autophagy, and induces Nrf2 nuclear translocation in PC12 cells

To verify the cytotoxicity of LPS towards PC12 cells, PC12 cells were treated with 0, 0.05, 0.10, 0.15, 0.20, 0.30, and 0.40 μg/mL LPS for 24 h. A CCK-8 assay revealed that there was no change in cell viability after treatment with LPS at 0.05, 0.10, 0.15, and 0.20 μg/mL concentrations, while LPS concentrations of 0.30 and 0.40 μg/mL led to a significant reduction in cell viability ($p < 0.05$; Fig. 3A). Therefore, 0.20 μg/mL LPS was used for further study. Next, we explored the effect of LPS on MALAT1 expression, autophagy, and Nrf2 nuclear translocation in PC12 cells. An RT-qPCR assay showed that MALAT1 expression was markedly up-regulated in the 0.20 μg/mL LPS group ($p < 0.001$; Fig. 3B). Additionally, LPS treatment greatly increased the nuclear Nrf2 (N-Nrf2) level and decreased the cytoplasmic Nrf2 (C-Nrf2) level ($p < 0.001$; Figs. 3C–3D). Besides, low-dose LPS pretreatment increased the expression levels of Beclin1 and LC3II/LC3I ($p < 0.001$; Figs. 3E, 3F). These data suggest that low-dose LPS promotes an increase in the expression of MALAT1, autophagy activation, and Nrf2 nuclear translocation in PC12 cells.

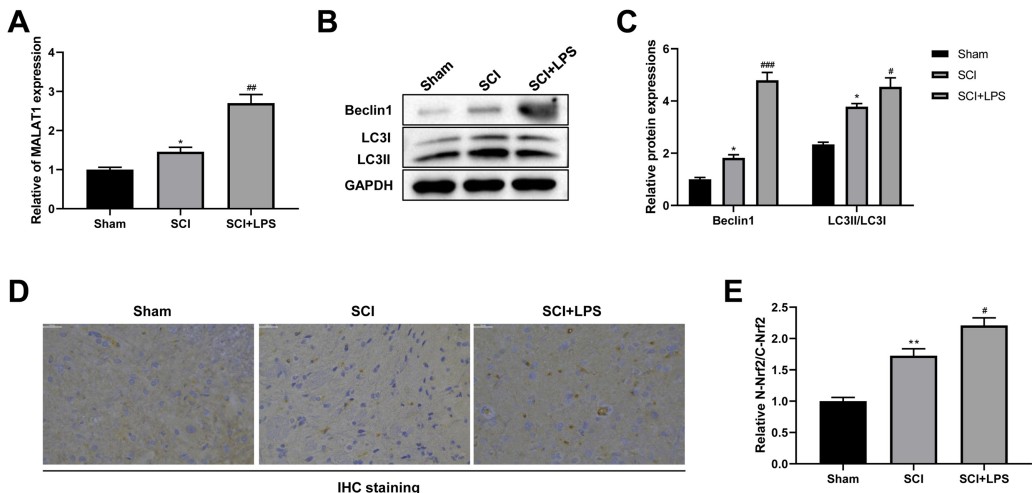

**Figure 2** **Low-dose LPS increases MALAT1 expression, activates autophagy, and induces Nrf2 nuclear translocation in the *in vivo* SCI model.** SD rats were intraperitoneally injected with 0.2 mg/kg LPS 72 h prior to SCI. (A) RT-qPCR assay was used to detect the MALAT1 expression in spinal cord tissues. (B, C) The protein expressions of autophagy-related proteins Beclin1 and LC3II/LC3I were analyzed by Western blot. (D, E) IHC staining was used to detect the expression of Nrf2 in the nucleus and cytoplasm in spinal cord tissues. Scale Bar = 50 μm. $n = 8$ each group. * $P < 0.05$ *vs*. Sham; # $P < 0.05$, ## $P < 0.01$, ### $P < 0.001$ *vs*. SCI.

## Low-dose LPS suppresses cell apoptosis, increases MALAT1 expression, activates autophagy, and promotes Nrf2 nuclear translocation in the *in vitro* SCI model

Next, we assessed the role of low-dose LPS on cell apoptosis in the *in vitro* SCI model. PC12 cells were pretreated with 0.20 μg/mL LPS for 24 h and then exposed to OGD/R to simulate the *in vitro* SCI model. Flow cytometric analysis revealed that the OGD/R group had a dramatically increased PC12 cell apoptosis compared to that in the control group ($p < 0.001$); LPS treatment suppressed the OGD/R-induced PC12 cell apoptosis ($p < 0.001$; Figs. 4A–4B). In addition, MALAT1 expression was markedly up-regulated in the OGD/R group compared to that in the control group ($p < 0.05$) and was further enhanced by LPS pretreatment ($p < 0.001$; Fig. 4C). Similarly, the expression levels of Beclin1 and LC3II/LC3I were significantly increased in the OGD/R group compared to those in the control group ($p < 0.05$); LPS administration further increased these expression levels ($p < 0.01$; Figs. 4D–4E). Moreover, LPS promoted Nrf2 translocation to the nucleus compared to the Nrf2 translocation level in the OGD/R group (Fig. 4F). These findings indicate that low-dose LPS has protective effects on OGD/R-induced PC12 cells and promotes an increase in MALAT1 expression, autophagy activation, and Nrf2 nuclear translocation in PC12 cells.

## Low-dose LPS inhibits cell apoptosis and promotes autophagy by upregulating MALAT1 expression in the *in vitro* SCI model

MALAT1 was knocked down in PC12 cells to explore its possible role in the SCI model *in vitro*. The transfection efficiency of siMALAT1 in PC12 cells was determined using an RT-qPCR assay ($p < 0.001$; Fig. 5A). We found that siMALAT1 dramatically suppressed

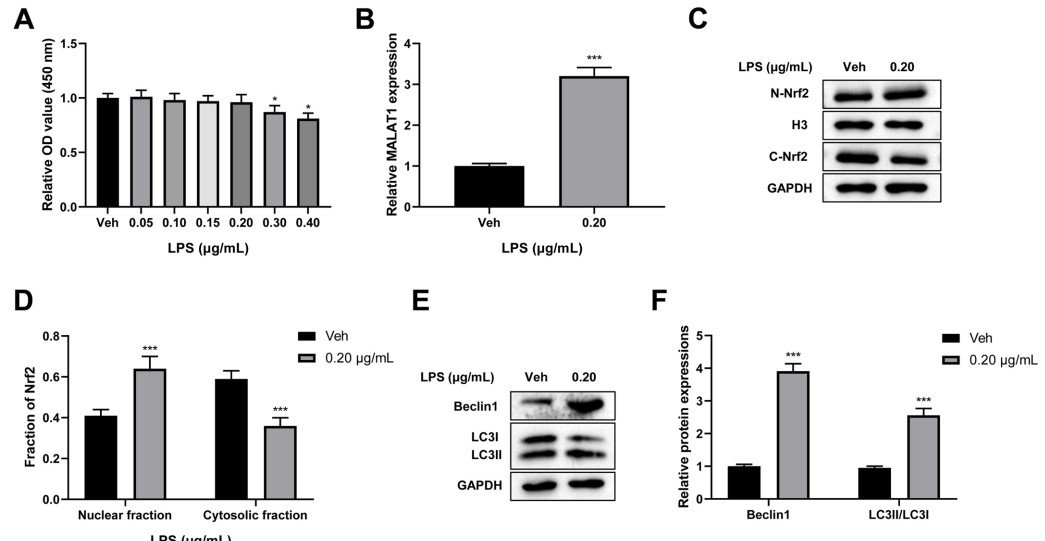

**Figure 3 Low-dose LPS increases MALAT1 expression, activates autophagy, and induces Nrf2 nuclear translocation in PC12 cells.** PC12 cells were treated with 0.00, 0.05, 0.10, 0.15, 0.20, 0.30, 0.40 $\mu$g/mL LPS for 24 h. (A) Cell viability was assessed by CCK-8 assay. (B) RT-qPCR assay was used to detect the MALAT1 level in 0.20 $\mu$g/mL LPS treated PC12 cells. (C, D) Nrf2 expression in the nucleus and cytoplasm was detected by Western blot. (E–F) Autophagy-related proteins Beclin1 and LC3II/LC3I were analyzed by Western blot. These experiments were repeated for three times. * $P < 0.05$, *** $P < 0.001$ *vs.* 0.00 $\mu$g/mL LPS.

the MALAT1 level in OGD/R-treated PC12 cells ($p < 0.01$), and LPS pretreatment further increased MALAT1 expression in these cells ($p < 0.01$; Fig. 5B). Flow cytometric analysis suggested that knockdown of MALAT1 promoted OGD/R-induced PC12 cell apoptosis ($p < 0.01$), which was largely reversed by LPS pretreatment ($p < 0.01$; Figs. 5C–5D). Furthermore, knockdown of MALAT1 suppressed the promotion of cell autophagy by OGD/R ($p < 0.01$), and LPS pretreatment reversed this effect ($p < 0.01$; Figs. 5E–5F). The above results indicate that MALAT1 is a vital factor for regulating the role of LPS in OGD/R-induced PC12 cell apoptosis and autophagy.

## MALAT1 inhibits cell apoptosis and promotes autophagy by promoting Nrf2 nuclear translocation in the *in vitro* SCI model

MALAT1 was overexpressed in PC12 cells, and the transfection efficiency was detected using an RT-qPCR assay ($p < 0.001$; Fig. 6A). To explore whether MALAT1 regulates autophagy in PC12 cells, PC12 cells were incubated with 3-MA for 24 h. Western blot suggested that, compared to the OGR/D group, MALAT1 overexpression greatly upregulated Beclin1 and LC3II/LC3I expression levels ($p < 0.01$), while 3-MA blocked this upregulation in OGR/D-treated PC12 cells ($p < 0.01$; Figs. 6B–6C). Western blot further showed that MALAT1 overexpression upregulated the expression of N-Nrf2 and downregulated the expression of C-Nrf2 ($p < 0.01$); 3-MA partly reversed the effects of MALAT1 overexpression ($p < 0.01$; Figs. 6D–6E). Flow cytometric analysis revealed that MALAT1 overexpression significantly

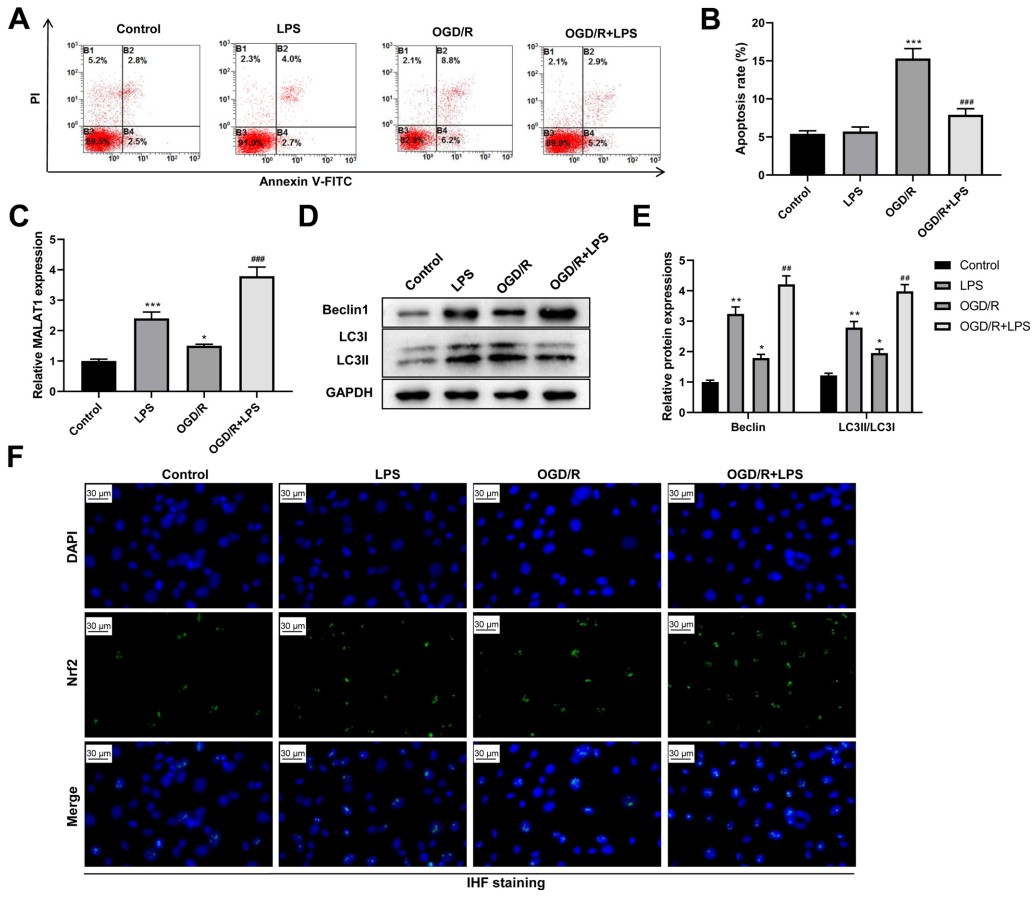

**Figure 4 Low-dose LPS suppresses cell apoptosis, increases MALAT1 expression, activates autophagy, and promotes Nrf2 nuclear translocation in the *in vitro* SCI model.** PC12 cells were pretreated with 0.20 μg/mL LPS for 24 h and then exposed to OGD/R to simulate the SCI model *in vitro*. (A, B) Flow cytometry assay was used to detect the apoptosis extent of PC12 cells. (C) RT-qPCR assay was performed to detect MALAT1 expression. (D, E) The protein expressions of autophagy-related proteins Beclin1 and LC3II/LC3I were analyzed by Western blot. (F) Nrf2 nuclear translocation in PC12 cells was observed by immunofluorescence staining. These experiments were repeated for three times. * $P < 0.05$, ** $P < 0.01$, *** $P < 0.001$ *vs.* Control; ## $P < 0.01$, ### $P < 0.001$ *vs.* OGD/R.

suppressed the OGD/R-induced PC12 cell apoptosis ($p < 0.01$) and that this effect was effectively reversed by 3-MA ($p < 0.01$; Figs. 6F–6G). These findings suggest that MALAT1 promotes Nrf2 nuclear translocation and autophagy activation and suppresses cell apoptosis in OGD/R-induced PC12 cells.

## Low-dose LPS alleviates neurologic dysfunctions and histopathological damage by upregulating MALAT1 in the *in vivo* SCI model

To determine the effect of MALAT1 on SCI *in vivo*, model rats were transfected with lentivirus particles encoding siRNAs for MALAT1 knockdown. LPS pretreatment promoted MALAT1 expression ($p < 0.01$), and MALAT1 knockdown significantly reversed this effect ($p < 0.001$; Fig. 7A). The rats in the LPS pretreatment group showed higher BBB scores

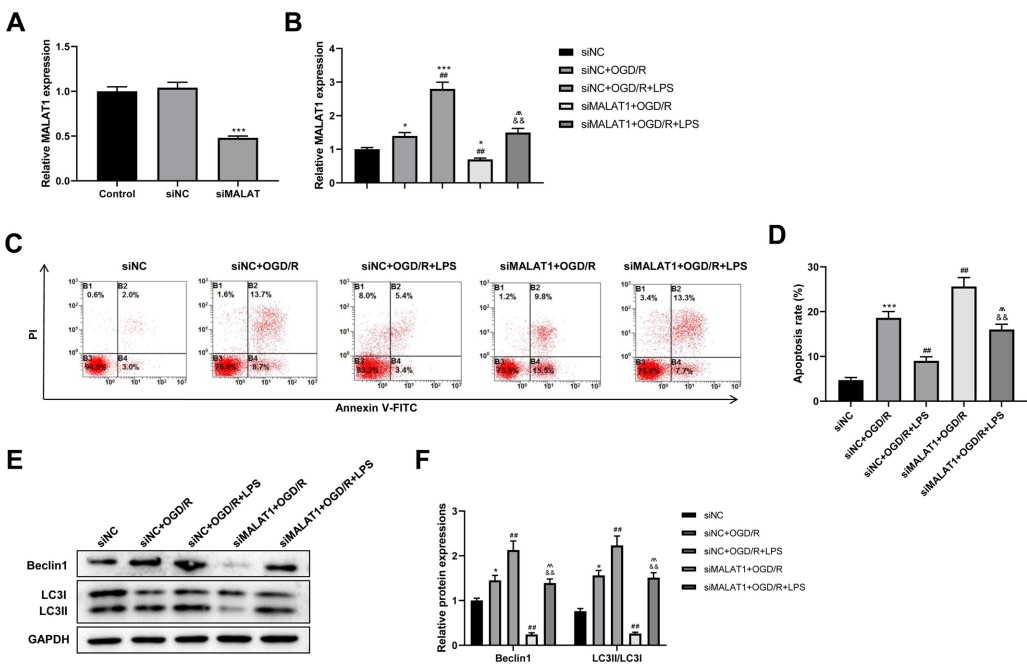

**Figure 5** **Low-dose LPS inhibits cell apoptosis and promotes autophagy by upregulating MALAT1 expression in the *in vitro* SCI model.** siMALAT1 or siNC was transfected to LPS and/or OGD/R-induced PC12 cells. (A) The transfection efficiency of siMALAT1 in PC12 cells was evidenced by RT-qPCR assay. (B) RT-qPCR assay was used to assess the MALAT1 level. (C, D) Flow cytometry assay was performed to detect the apoptosis extent of PC12 cell. (E, F) The protein expressions of autophagy-related proteins Beclin1 and LC3II/LC3I were analyzed by Western blot. These experiments were repeated for three times. * $P < 0.05$, *** $P < 0.001$ *vs.* siNC; ## $P < 0.01$ *vs.* siNC+OGD/R; ^ $P < 0.01$ *vs.* siNC+OGD/R+LPS; && $P < 0.01$ *vs.* siMALAT1+OGD/R.

than those in the SCI group ($p < 0.01$); MALAT1 knockdown decreased these BBB scores ($p < 0.01$; Fig. 7B). Furthermore, HE staining indicated that MALAT1 knockdown significantly reversed the inhibition of spinal cord tissue histopathological damage by LPS in the *in vivo* SCI rats (Fig. 7C). LPS pretreatment significantly reduced the cell apoptosis ratio compared to that in the SCI group ($p < 0.01$), while MALAT1 knockdown markedly reversed this effect ($p < 0.01$; Figs. 7D–7E). These results confirmed that low-dose LPS plays a protective role in SCI by promoting MALAT1 expression *in vivo*.

## Low-dose LPS activates autophagy and Nrf2 nuclear translocation by upregulating MALAT1 in the *in vivo* SCI model

We further assessed whether low-dose LPS influences autophagy activation and Nrf2 nuclear translocation in the *in vivo* SCI model upon MALAT1 overexpression. Western blot revealed that LPS pretreatment dramatically increased Beclin1, LC3II/LC3I, and N-Nrf2/C-Nrf2 levels ($p < 0.01$), while MALAT1 knockdown reversed these effects ($p < 0.01$; Figs. 8A–8D). These results suggest that LPS participates in autophagy activation and Nrf2 nuclear translocation by upregulating MALAT1 in the *in vivo* SCI model.

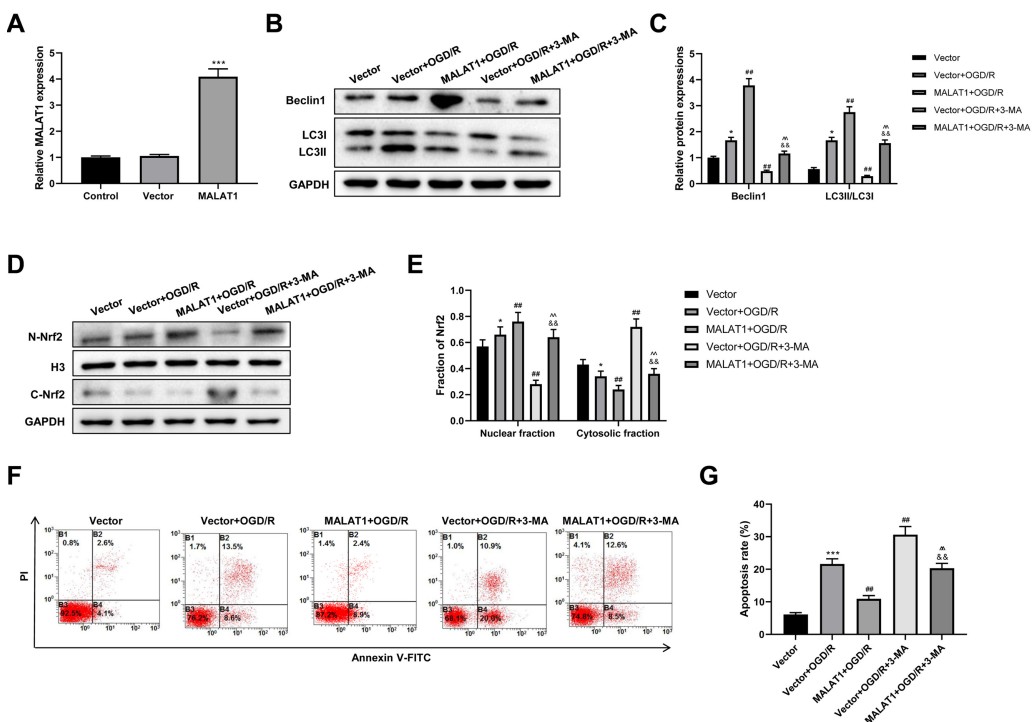

**Figure 6** **MALAT1 inhibits cell apoptosis and promotes autophagy by promoting Nrf2 nuclear translocation in the *in vitro* SCI model.** (A) MALAT was overexpressed in PC12 cells and the transfection efficiency was detected by RT-qPCR assay. MALAT was further overexpressed in OGD/R-induced PC12 cells, and 3-MA was incubated with PC12 cells for 24 h. (B, C) The protein expressions of autophagy-related proteins Beclin1 and LC3II/LC3I were analyzed by Western blot. (D, E) The protein expression of Nrf2 in the nucleus and cytoplasm of PC12 cells was detected by Western blot. (F, G) Flow cytometry assay was used to detect the apoptosis extent of PC12 cells. These experiments were repeated for three times. * $P < 0.05$, *** $P < 0.001$ *vs.* Vector; ## $P < 0.01$ *vs.* Vector+OGD/R; ^^$P < 0.01$ *vs.* MALAT1+OGD/R; && $P < 0.01$ *vs.* Vector+OGD/R+3-MA.

## DISCUSSION

In the study, we found that low-dose LPS had protective effects on nerve function and pathophysiology in SCI. More importantly, we found that probably through the activation of autophagy and the suppression of nerve cell apoptosis.

Inflammation is an important pathological process in SCI (*Tang et al., 2020*). Interestingly, recent studies have shown that treatment with low-dose LPS significantly enhances central nervous system tolerance and provides a long-term neuroprotective effect (*Lv et al., 2018*; *Wang et al., 2017*). For instance, low-dose LPS preconditioning attenuated cerebral ischemia/reperfusion injury via suppressing the inflammatory response and cell apoptosis (*Lv et al., 2018*). In an *in vivo* SCI rat model, low-dose LPS pretreatment exerted a neuroprotective effect (*Zhu et al., 2015*). In the present study, to explore the effect of low-dose LPS on SCI progression, both *in vitro* and *in vivo* SCI models were established. We found that low-dose LPS pretreatment suppressed neurologic dysfunctions and histopathological damage after SCI in rats. Moreover, we found that low-dose LPS

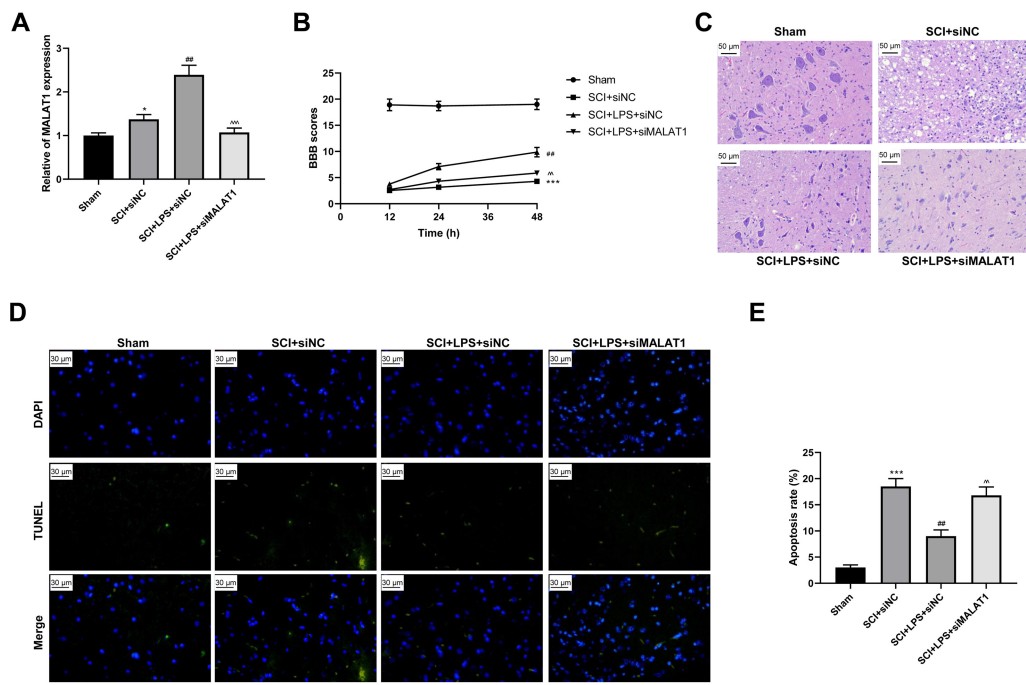

**Figure 7  Low-dose LPS alleviates neurologic dysfunctions and histopathological damage by upregulating MALAT1 in the *in vivo* SCI model.** SD rats were transfected with MALAT1 knocking down lentiviruses. (A) BBB scores were used to analyze the neurologic function after SCI. (B) The pathological and morphologic injury of spinal cord was observed by HE staining. Scale Bar = 50 μm. (C, D) TUNEL assay was used to assess the cell apoptosis in spinal cord tissues. Scale Bar = 50 μm. $n = 12$ each group. *** $P < 0.001$ *vs.* Sham; ## $P < 0.01$ *vs.* SCI+siNC; ^^$P < 0.01$ *vs.* SCI+LPS+siNC.

inhibited cell apoptosis after SCI in PC12 cells by flow cytometry. These findings indicate that low-dose LPS has protective effects on SCI and are consistent with the findings of a previous study by *Zhu et al. (2015)*.

MALAT1 is one of the most extensively studied lncRNAs (*Zhang, Hamblin & Yin, 2017a*). It was reported to be highly abundant in brain tissues, especially in the high-activity areas of the human neocortex. Several studies have suggested that MALAT1 might be involved in neural development and functions (*Liu et al., 2022*; *Was et al., 2022*). MALAT1 was also reported to participate in SCI development by regulating the inflammatory response of microglia (*Zhou et al., 2018*). Moreover, one study found that MALAT1 exerted a neuroprotective effect in a rat model of SCI by regulating miR-204 (*Qiao et al., 2018*). Another study reported that LPS treatment promoted MALAT1 expression in human pulmonary alveolar epithelial cells (*Nan et al., 2020*). In the present study, we found that LPS pretreatment upregulated MALAT1 expression in an *in vivo* SCI rat model and an *in vitro* SCI PC12 cell model.

LPS treatment can activate autophagy. *Sul et al. (2017)* found that LPS induces osteoclast autophagy as well as osteoclastogenesis *in vitro*. Moreover, *Li et al. (2018)* revealed that autophagy was activated in LPS-injured PC12 cells and SCI rats. Other studies have shown that MALAT1 could relieve cell injuries, such as brain microvascular endothelial cell

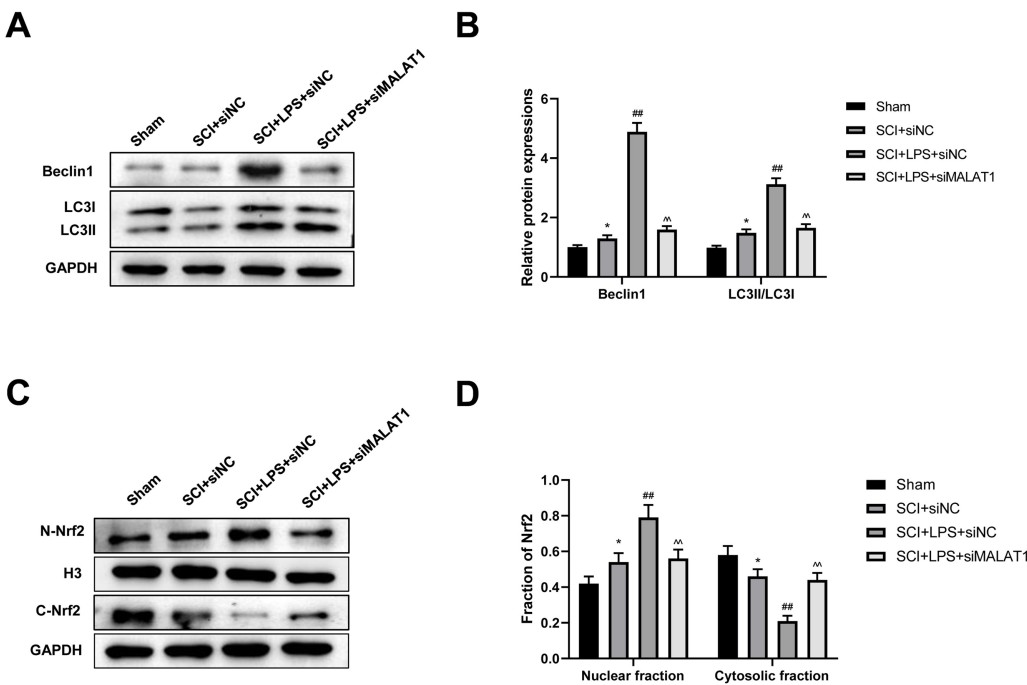

**Figure 8  Low-dose LPS activates autophagy and Nrf2 nuclear translocation by upregulating MALAT1 in the *in vivo* SCI model.** SD rats were transfected with MALAT1 knocking down lentiviruses. (A, B) The protein expressions of autophagy-related proteins Beclin1 and LC3II/LC3I were analyzed by Western blot. (C, D) Western blot was used to detect the protein expression of Nrf2 in the nucleus and cytoplasm of spinal cord tissues. $n = 12$ each group. * $P < 0.05$ *vs.* Sham; ## $P < 0.01$ *vs.* SCI+siNC; ^^$P < 0.01$ *vs.* SCI+LPS+siNC.

injury, by promoting autophagy (*Li, Li & Tang, 2017*). Interestingly, some studies have also reported that MALAT1 had a negative effect on cell injury through autophagy. For example, one study showed that MALAT1 transfection induced cardiomyocyte injury by increasing Beclin1 expression (*Wang et al., 2019b*). In the present study, we found that LPS treatment dramatically enhanced the expression levels of the autophagy-related proteins Beclin1 and LC3II/LC3I in both *in vitro* and *in vivo* SCI models.

An increase in the autophagy flux has been confirmed to inhibit apoptosis and play a neuroprotective role in SCI by numerous studies. For instance, rapamycin was found to exert a neuroprotective effect to promote SCI recovery in rats by enhancing autophagy (*Li et al., 2019*). *Rong et al. (2019)* found that exosomes derived from neural stem cells suppressed cell apoptosis and the neuroinflammatory response after SCI by activating autophagy. *Zhao et al. (2017)* suggested that resveratrol can protect SCI via the activation of autophagy and the inhibition of apoptosis. Therefore, we speculated that low-dose LPS might be involved in SCI through MALAT1-mediated autophagy regulation. In support of this speculation, the present study showed that low-dose LPS inhibited cell apoptosis and promoted autophagy by increasing MALAT1 expression in *in vitro* and *in vivo* SCI models.

Nrf2 is a key transcriptional regulator in important cellular defense and antioxidant stress pathways (*Bouvier et al., 2017*). Nrf2 can regulate antioxidant, anti-inflammatory,

and other cell protection-related genes and proteins to enhance the ability of the body to resist injuries, such as those resulting from oxidative stress (*Galiè et al., 2019*; *Wang et al., 2020*). Dysregulation of autophagy was shown to result in prolonged Nrf2 activation (*Jiang et al., 2015*). It was also reported that Nrf2 modulates autophagy, thereby alleviating oxidative stress and the inflammatory response (*Zhao et al., 2020*; *Zuo et al., 2019*). A study has also shown that MALAT1 suppressed umbilical vein endothelial cell injury by increasing the expression of Nrf2 pathway components (*Zeng et al., 2018*). Another study reported that LPS improved neurological function via the activation of Nrf2 after SCI (*Li et al., 2016*). In the present study, we further revealed the enhancement of Nrf2 nuclear translocation upon LPS pretreatment and MALAT1 overexpression. Based on these findings, we believe that low-dose LPS may induce MALAT1 upregulation to promote autophagy and activate Nrf2 nuclear translocation, thereby regulating the expression of apoptosis-related genes and inhibiting apoptosis of nerve cells to alleviate SCI.

## CONCLUSION

Overall, the present study showed that pretreatment with low-dose LPS significantly alleviated SCI-induced neuro-apoptosis and autophagy activation via upregulating MALAT1 and promoting Nrf2 nuclear translocation. These findings indicate that low-dose LPS has protective effects on SCI via the MALAT1/Nrf2 axis, providing a potential novel strategy for the treatment of SCI.

### Funding

This study was supported by the National Natural Science Foundation of China (Grant No: 82260257), Key Research Project of Yunnan Provincial Science and Technology (Grant No.: 202102AA310042; 202001AS070028), Yunnan Health Training Project of High Level Talents (Grant No: H-2018100), Key project of Yunnan clinical medicine research center (Grant No: 2022YJZX-GK02) and Training and Development Project of Yunnan clinical medicine center (Grant No.: 2021LCZXXF-CS02). The funders had no role in study design, data collection and analysis, decision to publish, or preparation of the manuscript.

### Grant Disclosures

The following grant information was disclosed by the authors:
The National Natural Science Foundation of China: 82260257.
Key Research Project of Yunnan Provincial Science and Technology: 202102AA310042, 202001AS070028.
Yunnan Health Training Project of High Level Talents: H-2018100.
Key project of Yunnan clinical medicine research center: 2022YJZX-GK02.
Training and Development Project of Yunnan clinical medicine center: 2021LCZXXF-CS02.

### Competing Interests

The authors declare that there are no competing interests.

## Author Contributions

- Jianhua Hu performed the experiments, prepared figures and/or tables, authored or reviewed drafts of the article, and approved the final draft.
- Kun Huang performed the experiments, prepared figures and/or tables, authored or reviewed drafts of the article, and approved the final draft.
- Feilong Bao analyzed the data, prepared figures and/or tables, and approved the final draft.
- Shixiao Zhong analyzed the data, prepared figures and/or tables, and approved the final draft.
- Qianbo Fan analyzed the data, prepared figures and/or tables, and approved the final draft.
- Weichao Li conceived and designed the experiments, prepared figures and/or tables, authored or reviewed drafts of the article, and approved the final draft.

## Animal Ethics

The following information was supplied relating to ethical approvals (i.e., approving body and any reference numbers):

The First People's Hospital of Yunnan province, Affiliated Hospital of Kunming University of Science and Technology.

## Data Deposition

The raw data is available in the Supplemental File.

## Supplemental Information

Supplemental information for this article can be found online at http://dx.doi.org/10.7717/peerj.15919#supplemental-information.

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
