# Peer review of "Low-dose lipopolysaccharide inhibits spinal cord injury-induced neuronal apoptosis by regulating autophagy through the lncRNA MALAT1/Nrf2 axis"

_PeerJ, doi:10.7717/peerj.15919_

## Round 0.1 · original submission · Major Revisions

Please address the reviewer's comments in a point-to-point response letter and engage the services of an English editing service.

Reviewer 1 ·

Basic reporting

To authors,
This study sought to investigate the involvement of the MALAT1/Nrf2 axis in LPS-induced protection against Spinal cord injury (SCI) by manipulating the expression of MALAT1 in vivo and in vitro. The authors observed that the LPS-induced protection against SCI was attenuated by MALAT1 silencing, but enhanced by MALAT1 overexpression, which is accompanied by the corresponding translocation of Nrf2 into the nucleus and autophagic induction. Upon these data, the authors conclude that low-dose LPS pretreatment protects against SCI by initiating autophagy via the lncRNA MALAT1/Nrf2 axis. This study is interesting and provides valuable knowledge for our understanding of the etiology of SCI. The reviewer has some comments as follows.

1. In the abstract section, the methods are relatively long, and the results do not mention the MALAT1 manipulation, two key points for this study; In Figure 1, Panel C is the quantification of Panel D. It is better to organize them as one panel with placing the typical images before the quantification. The same changes should be made to Figure 2 B and C, Figure 2 D and E, Figure 3 C and D, Figure 3 E and F, Figure 4 A and B, Figure 4 D and E, Figure 5 D and D, Figure 5 E and F, Figure 6 B and C, Figure 6 D and E, Figure 6 F and G, Figure 7 C and D, Figure 8 B and C, Figure 8 D and E.

2. Upon H&E staining, it is questionable to detect inflammatory cell infiltration. It is better to delete the “…as well as inflammatory cell infiltration…”. In Figure 2 D and E, it is better to add information for staining, and additional inserts with higher magnification showing the translocation of Nrf2 into the nucleus. The same revision for Figure 4F is required.

3. Place “nuclear fraction” before or after the western blots of N-Nrf2 and H3, and “cytosolic fraction” before or after the western blots of C-Nrf2 and GAPDH in Figures 3D, 6E, and 8E.

4. Figure 7 needs the data to support the efficiency of MALAT1 knocking down.

5. In Figure 3, change “0.00” to “Vehicle/Veh”.

6. There is no data to support the Nrf2 nuclear translocation to contribute to LPS-induced protection. The declaration of “these protective effects were attributable to … and Nrf2 nuclear translocation,” is better to be deleted.

7. In the discussion section, the second paragraph is somewhat beyond the current study. It is better to shorten this paragraph largely. The description of Lines 374 to 376 is better to be revised because there is no data of the expression of apoptosis-related genes in this study.

Experimental design

The experimental design is good and meets the PeeJ's standards.

Validity of the findings

This study is interesting, and the conclusion is basically supported by data.

Additional comments

The writing is somewhat unconcise.

Reviewer 2 ·

Basic reporting

"Low-dose lipopolysaccharide inhibits spinal cord injury-induced neuronal apoptosis by regulating autophagy through LncRNA MALAT1/Nrf2 axis" declares that Low-dose lipopolysaccharide has protective roles on SCI through upregulating MALAT1 and the activation of autophagy, leading to promote Nrf2 nuclear translocation and suppress nerve cell apoptosis. It is significance for the therapy for SCI patients. But there are some problems must be solved:
1. Why do you choose 0.2 mg/kg LPS for SD rats and choose 24 hours for PC12 cells to be pretreated with LPS?
2. The scale bar in the Figure 1B is not clear.
3. In the results of TUNEL staining, please show the images of single DAPI staining and TUNEL staining in Figure 1D and Figure 7D.
4. In the results of Immunofluorescence staining, please show the images of single DAPI staining and Nrf2 staining in Figure 4F.
5. The orders of many figures must be rearranged. Such as: Figure 2B and Figure 2C, Figure 2C must be showed before Figure 2B. Another: Figure 2D and 2E, Figure 3C and 3D, Figure 4A and 4B, Figure 4D and 4E, Figure 5C and 5D, Figure 5E and 5F, Figure 6B and 6C, Figure 6D and 6E.
6. Page 6: "Reverse transcription-quantitative PCR (RT-qPCR)"...The grammar of the sentence "RT-qPCR was implemented using were performed by ABI Prism 7500 PCR system (AppliedBiosystems, USA) utilizing SYBR Green Master Mix (Takara, Japan)." must be checked.

Experimental design

no comment

Validity of the findings

no comment

Reviewer 3 ·

Basic reporting

I don't think the author's experiments can prove that the protective effect of low-dose LPS is related to autophagy and the pathways it does.

Experimental design

The reasons are as follows, 1. LPS has traditionally been more closely associated with inflammation than autophagy. 2. The classic action pathway of LPS is TLR4. 3. The author chose autophagy as the core mechanism without excluding the more core inflammatory effect of LPS, and indeed chose the anti-oxidation pathway of nrf2, the logic is difficult to understand. 4. Low dose is a relative concept, how low can be considered low, and inflammation also has a protective effect in the early stage. According to the role of LPS, I can actually think that this is the protective effect of inflammation.

Validity of the findings

Based on the most basic experimental design and logic, I think this is unreasonable and difficult to modify. It is recommended to reject the manuscript.

Additional comments

NONE

---

## Round 0.2 · accepted · Accept

Reviewers have completed their review.

The Figure legends need to explain the number of rats used (biological) replicates and whether there were technical replicates. Please address this during the proof stage.

Reviewer 1 ·

Basic reporting

No more comments.

Experimental design

No more comments.

Validity of the findings

No more comments.

Additional comments

The authors answered all my concerns and revised the manuscript. The reviewer has no more comments.

Reviewer 2 ·

Basic reporting

The experiments are well designed, carefully executed, thoroughly analyzed and clearly written. All comments have been well responded and revised. I agree to accept.

Experimental design

no comment

Validity of the findings

no comment

Reviewer 3 ·

Basic reporting

Although I still reserve my personal opinion academically, after the author's reply, I think this article can be accepted, and different academic voices are worth discussing and publishing.
Wish you all the best.

Experimental design

How the low-dose LPS determined in the in vivo and in vitro experiments is converted is worthy of caution

Validity of the findings

Interesting, worthy of in-depth study

Additional comments

NA